

**3-Dimensional modeling of 2014-Malin Landslide, Maharashtra using satellite-derived data: A**
**quantitative approach to numerical simulation technique**
Shovan Lal Chattoraj[1], Prashant K. Champati ray[1], Sudhakar Pardeshi[2], Vikram Gupta[3], Yateesh
Ketholia[1]
**Affiliation:**[1]Indian Institute of Remote Sensing, 4-Kalidas Road, Dehradun-248001
[2]Univeristy of Pune, Ganeshkhind, Pune, Maharashtra-411007
[3]Wadia Institute of Himalayan Geology, 33, GMS Road. Dehradun-248001
**Corresponding Author:**
Shovan Lal Chattoraj
Email: shovan.iitb@gmail.com
Phone: 0135-2524157
**Abstract:**
Debris flows, a type of landslides, are not nowadays limited only to the periodic devastation of the
geologically fragile Himalaya but also ubiquitous in weathered Deccan Volcanic Province of the
cratonic south Indian peninsula. Comprehensive assessment of landslide hazard, pertinently, requires
process-based modeling using simulation methods. Development of precipitation triggered debris
flow simulation models of real events are still at a young stage in India, albeit, especially in
tectonically less disturbed regions. A highly objective simulation technique has therefore been
envisaged herein to model the debris flow run-out happened in Malin. This takes cues from a high-
resolution DEM and other ancillary ground data including geotechnical and frictional parameters. The
algorithm is based on Voellmy frictional (dry and turbulent frictional coefficients, μ and ξ
respectively) parameters of debris flow with pre-defined release area identified on high-resolution
satellite images like LISS-IV and Cartosat-1. The model provides critical quantitative information on
flow 1) Velocity, 2) Height, 3) Momentum, and 4) Pressure along the entrainment path. The
simulated velocity of about 16m/s at mid-way the slide plummeted to 6.2 m/s at the base with
intermittently increased and decreased values. The simulated maximum height was 3.9m which
gradually declined to 1.5m near the bottom. The results can be beneficial in engineering intervention
like the construction of check dams to digest the initial thrust of the flow and other remedial measures
designed for vulnerable slope protection.
**Key Words**: landslide, debris flow, Malin, simulation, satellite image.



## 1. Introduction:

Mass wasting - a general term for all kind of movements, has become a treacherous issue in the Himalaya. Quite frequently, especially during rainy season, landslides are witnessed in Lesser and Central Himalaya causing severe loss to man and property. Moreover, these landslides may lead to some critical problems such as blockade of rivers, which may incite secondary catastrophic disaster such as floods, as was the case in 2013-Kedarnath Tragedy in the Uttarakhand Himalaya. Off late, Sahyadri hills in the Western Ghats have witnessed many landslides, particularly during the rainy season, causing severe loss to humankind and property (Gujarathi and Mane, 2015).

Considering the graveness of the issue, many researchers and experts have analyzed landslides from all perspectives, i.e., to model, predict or to design preventive measures. Subsequently, the number of well-tested and documented empirical methods have been evolved to determine dynamic and kinematic parameters of the flow. However, some numerical simulation techniques are more preferred to predict flow paths and characterize the entrainment process (Tsai et al. 2011; Quan Luna et al. 2011, Evans et al. 2011). The underlying principle of such events can be applied to a variety of processes including snow avalanche. Debris flows, landslides, mudflows and even rock falls and has therefore found to be significant in disaster management. Although well tested empirical methods adopted by Heim 1932, Scheidegger 1973, Corominas, 1996, Nicolettiand Sorriso-Valvo, 1991, Li, 1983, Hungr, 1995 are available to determine dynamic characteristics of a flow, numerical simulation techniques, such as Hungr, 2006, Iverson, 1997, Savage and Hutter, 1989, Chen and Lee ,2000, Iverson and Delinger, 2001, McDougall and Hungr, 2004, Sousa and Voight, 1991, Hungr, 1995 (DAN), Volellmy, 1955, Hungr and Evans,(DAN 3D ), 1996, 2004, Hungr and McDougall,2009 are now being widely applied to predict flow paths and characterize the entrainment process.

Pertinently, as for the Indian subcontinent, the Himalayan region has experienced many devastating landslides in the past. Most of the landslides in the Himalayan region have a major debris flow component that travels some distance causing enormous damage enroute (Chattoraj, 2016, Chattoraj and Champati ray 2015; Champati and Chattoraj, 2014). On the contrary, debris flows are less abundant in Western Ghats. However, most of the works mentioned above, reports either the geo-engineering aspects of landslides or hazard/ susceptibility mapping leading to damage assessment. Comprehensive assessment of landslide hazard which requires process based modeling using numerical simulation methods is still lacking or at nascent stage in Indian subcontinent as a whole.





Precipitation triggered debris flow models have, albeit, been attempted in similar tectonically disturbed regions of the world and holds tremendous opportunity in implementation of a successful strategy for landslide hazard mitigation (Brand 1995, Champati ray et al. 2013, Deganutti et al. 2000; Hungr et al. 1987; Scott 2000). The present study aims to fill this knowledge gap by focusing on numerical analysis of major landslides/debris flow movements and simulate landslides that occurred in the Western Ghats. This study leads to derivation of the important physical flow parameters taking cues from Earth Observation techniques to understand the root cause of the devastation, which is essential for effective mitigation measures.

In the present study, RAMMS (Rapid Mass Movements Software) developed by WSL Institute of Snow and Avalanche, Switzerland has been used, which is a state-of-the-art numerical simulation model that predicts the motion of a naturally occurring mass from a head (release area) to base (deposition area) in three dimensions. The present study aims to address landslides/debris flow movement and simulate the landslide event that had occurred in the Malin area, the northern part of the Sahyadri hill, in the wee hours of 30[th] July 2014 following torrential rainfall. It engulfed 40 houses and gobbled up 151 people as per. The event was classified as an unchannelized debris flow consisting mainly of semi-consolidated, basalt-derived, silt to coarse sand-sized, poorly sorted soil, highly saturated with water which was triggered by intense monsoonal precipitation on leeward side of a slope underlain by thick alternating basaltic layers of varied composition and physical characteristics (Champati ray and Pardeshi, 2014).

The outputs of such simulated flows are likely to provide the stake holders actual insight of the cause of these events and associated disasters. Extensive landslide mapping at large scales complimented by this kind of 3-dimensional modeling of landslides will provide adequate information to understand the event and plan for the mitigation measures in future (Champati ray et al. 2013; Herva´set et al. 2003).

## 2. Study area and location

Malin village is located at latitude 19⁰09'40.84'' N and longitude 73⁰41'18.41'' E from 775m (avg.) above MSL (SOI Toposheet no. E43B/12) on a southeasterly facing slope of a small valley oriented along the NNE-SSW direction (Fig. 1). Downhill Malin village, a streamlet flows in SE direction which meets Bubranadi, a tributary of Ghod river, which in turn becomes contributory to Bhima



river. The Bhima river system forms part of the Upper Godavari basin. The Ghod river is dammed at Ambegaon forming the Dimbhe reservoir. This reservoir is fed by two significant inlets, the northern one of which flows close to Malin. Besides the main Dimbhe dam, there is a small dam at 9 km upstream on the Bubranadi. The upstream tail end limit of this reservoir water stops at about 1km away from Malin village in the upstream direction.

## 3. Regional Geology and Geomorphology:

Geologically, the Malin and adjoining area are embedded/overlain by Deccan Volcanic Province (DVP) of peninsular India consisting of numerous horizontal to gently dipping/inclined lava flows. The flows are characteristically transacted by linear discontinuities like parallel joints and fractures which are revealed (or reflected) in the form of lineaments and drainage systems have developed (along these discontinuities). The major trend of the lineaments are observed to be NW-SE and NNE/NE – SSW/SE directions (Champati ray and Pardeshi, 2014; Ramaswami et al. 2015). GSI, 1995 has defined three types of lava flows viz.1) fine-grained aphyric pahoehoe flows (Karla Formation), 2) Aphyric to sparsely phyric flows and Megacryst flow(Indrayani Formation), 3) fine to medium grained aphyric flows (Upper Ratnagarh Formation). These formations, in total, accommodates14 flows (Champati ray and Pardeshi, 2014).

In this area, the Sahyadri range is divided into two parts viz.1) high hills and adjoining plains located in the western part and 2) denudational hills and associated river valleys (Ghod and Bhima river)in the eastern part (Fig. 2). The study area falls in the second part. However, both the hill ranges show extensive plateau development owing to horizontal nature of lava flows. The small valley near Malin is located at an elevation of 680m, the village itself at 700-710m, followed by terrace at 750m, 800 and 840m on Cartosat-1 stereo-pair derived DEM. On SRTM DEM, the valley is located at 750m, Malin village at 770m, the terrace at 827 and 940m. Overall the relief difference is around 160-180m from the valley bottom to hilltop with an average slope of 11-13$^0$, and on the steepest section, the slope is 21$^0$.

## 4. Methodology and input data

Multi-temporal and multi-resolution Earth Observation satellite data products and derived information have been used to set parameters for flow modeling (table 1). Flow modeling has been developed and validated against the actual events of 2014 by ground checking.



### 4.1 Satellite Data used:

Indian Remote Sensing Satellite data products such as LISS-IV (Resourcesat 2) data sets acquired on 8[th] January, and Cartosat-1 data acquired on 3[rd] March 2011 were analyzed mainly for pre-event analysis (Table 1). Post-event changes were compared using LISS-IV (1[st] Feb 2015) and Cartisat-1 (6[th] April 2015). DEM (Res. 10m) was generated using pre-event Cartsat-1 stereo-pair in LPS module of Erdas Imagine software (v. 2014). Ancillary Earth observation data like SPOT images of Google Earth and terrain information derived from SRTM DEM Version 4 were also referred as detailed in table1.

### 4.2 Debris flow run-out modeling

The essential dataset required for the physically based model are topographic data (digital elevation model), release area and release mass as well as information on friction for dry and liquid phases and geo-engineering parameters like an internal shear angle and density. Topographical data sets in the form of high-resolution digital elevation model (DEM) and the location of release area are the two most important parameters for flow modeling. DEM in the form of the ESRI ASCII Grid and ASCII X, Y, Z format is required for implementation.

Debris flow modeling for unchannelized flows (as observed in the present case) requires a known release area with a given initial height for block release (Rickenmann D 1999, Rickenmann, 2005; Rickenmann et al. 2006). Therefore, the release areas for debris flows have been identified using high-resolution satellite images (Cartosat-1 and LISS-IV) and derived DEM. The initiation zone in the study area is steeper with slope angle ranging between 30-70° with height varying from 925m to 765m. The depth of the initiation zone (depletion zone) varies from 1m to 1.2m (Fig. 3,4). The field observations revealed that the modeled landslide was initiated with weathered basalt derivatives/ debris and when it hit Malin village width of the slide was maximum (~150m).

### 4.3 Frictional parameters

The RAMMS numerical simulation model is based on rheological characters of the slope derived from shear strength parameters of the slope. This model divides the frictional resistance into two parts: a dry-Coulomb type friction (coefficient, $\mu$) and a velocity-squared drag or viscous-turbulent friction (coefficient, $\xi$). The frictional resistance S (Pa) is then defined as:

$$S= \mu \rho H g \cos(\varphi) + (\rho g U^2)/\xi$$





Where ρ is the density, g the gravitational acceleration, φ the slope angle, H the flow height and U the
flow velocity (Salm et al. 1990). The two major frictional input parameters are μ and ξ. However, it is
known from a law of friction that μ= tan φ, where φ is an angle of internal resistance that can be
determined in the laboratory. In the present case, direct shear test instrument was used to determine c
(cohesion) and angle of internal friction from soil samples collected from the study area.
The main difficulty in case of debris flow simulation is the much larger variety of debris flow
materials, which influence the choice of the friction parameters. RAMMS Debris Flow uses a single-
phase model, and it cannot distinguish between fluid and solid phases, and the entire mass is modeled
as a bulk flow. Therefore, the friction parameters should be varied to match the observed flow paths
in case of known debris flow events. It is quite possible that different events in the same torrent may
show differences in composition. This fact makes the calibration of the friction parameters much
more difficult. Therefore, numbers of simulations with different values for dry and viscous turbulent
frictional coefficients were carried so that there is a close match between the modeled flow run out
and actual field/ satellite photograph observations.  The results were validated with field data, and the
best-fitted simulation outputs were adopted for final analysis (Sosio et al. 2008).
Thus, some simulations were considered using various possible ranges of friction parameters. To find
the optimal friction values, a range of values were used. The range of dry friction ranges from 0.05 to
0.5 and for viscous turbulent flow is 100-800 m/s$^2$ (Sosio et al. 2008). Meanwhile other input
parameters viz. density of materials, release height, earth pressure coefficient (lambda) and the
percent of momentum were kept constant. Afterward, validation of simulation outputs was done
comparing the total length of run-out distance, and the aerial extent of run out vis-a-vis the actual
flow paths on the ground.
When the simulated flow spatially matched approximately 97% (pixel-wise) with real event, model
parameters were frozen at μ (Mu) = 0.49, ζ (Xi) = 460 m/s$^2$ and cohesion (c) value of 100kPa. For dry
friction value, it was observed from that an increase in the friction coefficient $\mu$ (Mu) causes a
decrease in the run-out distance due to increase in the basal friction of the flow. On the other hand,
the value of ζ (Xi) changes did not affect the run-out distance significantly. However, in general case,
an increase in ζ (Xi) value increases the run-out distance and results in a relatively smoother flow.
Amongst RAMMS model outputs, momentum is not absolute as it simply considers momentum as a
product of flow height and velocity. Thus the unit is m²/s. To get real momentum in (kg*m/s), this





value is multiplied by the density of debris and area under consideration. Additionally, this numeral
simulation model does not include 1) en-route erosion and 2) side channel contribution to the main
flowing mass along run out. In most of the cases, variation in output geophysical parameters is
reported due to above reason. Therefore, maximum valuation of parameters has been provided with
error values. The outputs bound within error limits ensure that run out is restricted to the real debris
flow channel as verified in the field and/or satellite image.

**5. Instrumental validation of Shear strength parameters**
RAMMS numerical simulation derived models require cohesion (c) and the frictional coefficient for
dry and liquid phases ($\mu$ and $\xi$ respectively) for soil/ debris as inputs. Cohesion is independent of
stress systems and is dependent more on geochemical properties of the material. Frictional coefficient
(static) for dry debris phase ($\mu$) is related to the topographic slope by the rule of friction: tan $\varphi$ = $\mu$
(considering the angle of sliding equal to the angle of repose).  Thus, theoretically, the instrument
derived and modeled inputs of shear strength parameters of a successful simulation should match, if
assumptions are within the error range. Direct shear instrument was utilized to measure cohesion (c)
and angle of internal resistance ($\varphi$) assuming prevailing maximum in-situ saturation level. The
outputs of the direct shear instrument were plotted in the bivariate plot using Mohr-Coulomb
equation, i.e., $\tau = \sigma$ tan $\varphi$ + c which is a straight line equation between normal and shear stress plot.
As each model is frozen once it approximates the real debris flow and its $\mu$ and $\varphi$ are cross-checked
with the instrumentally derived c and $\varphi$ values from the soil sample. It is to be noted that when shear
strength model inputs in RAMMS model and instrument derived outputs are comparable, then it is
considered that simulation model validates well with the real world situation.
The representative samples collected from the base of the flows were analyzed in electronic direct
shear testing equipment (Model No. AIM 104 (2kN), Make Aimil Ltd, New Delhi) at Indian Institute
of Remote Sensing, Dehradun at different saturation levels. Samples were tested at 0.25, 0.50 and 1
kgf/cm$^2$ normal load and consequent shear strength parameters at failure was calculated. The input
dry coefficient of friction fed in the model was thus was further crosschecked instrumentally. The
Mohr-Coulomb equation revealed that the cohesion (c) and angle of internal shear resistance ($\varphi$) of
semi-consolidated debris which is 98-116 KPa and 25-32° (i.e. $\mu$= 0.4 to 0.6) respectively which are
at par with modeled inputs.



## 6. Results and Discussion:

The debris flow reached the maximum height of approximately 3.9m near the release area (Fig. 5). It consistently decreased to 1 meter with slide's propagation. However, the height suddenly rose around the toe of the slide, probably to conserve momentum. The maximum velocity of about 16 m/s was attained somewhere mid-way the slide. The velocity profile of the slide is zigzag with fluctuating velocities. The velocity near release area was 10 m/s, which intermittently increased and decreased during the entire sliding event. The velocity at toe modeled to be to be 5-6 m/s-sufficient enough to bury a village! The sliding mass had maximum momentum in the lower half of the profile probably due to the attainment of maximum velocity mid-way. The value of momentum near the release area was around 8-9 $m^2/s$, which then decreased and again increased to a maximum of 26 $m^2/s$ and then gradually dwindled down to rest(Fig. 5).The pressure more or less followed the footprint of velocity with fluctuating values throughout the landslide event. Henceforth, the maximum value of 440 KPa was reached somewhere near the middle(Fig. 5).

This work enhanced the understanding of numerical models by studying their resemblance with real landslide/debris flow that contributed to the unprecedented disaster in Malin. The vital output parameters viz. velocity, height, momentum, and pressure can be used to provide insight of the event and extent of runout zone of future potential flows which also helps in the understanding of slope stability. Thus, this work bespeaks that numerical simulation modeling is capable of emulating natural events and outputs can be used for mitigation measures. The results can be very useful in engineering intervention like a construction of check dams to digest the initial thrust of the flow and other remedial measures designed for vulnerable slope protection. Integrated with extensive landslide mapping, 3-dimensional modeling of landslides will complimentarily provide the stakeholders actual insight of the cause of this type of event vis-à-vis its effective corrective measure. The model has not only produced reliable simulation results but also established the efficacy and versatility in application of models in a wide range of mass wasting events about different causative factors.

## 7. Conclusion:

Three-dimensional modeling of natural debris flow events by the satellite image-based analysis provided two most important results. First of all, the study provided a successful simulation of selected debris flow events and generated output parameters such as velocity, height, pressure, and





momentum taking inputs from remotely sensed and ancillary earth observation data products.
Secondly, it provided critical insight into the events and their consequences. Based on the study, it is
concluded that the modeled flows have provided debris with sufficient height, velocity, and
momentum that devastated the whole area. The maximum height of the debris has been revealed to
approximately 4m which along entrainment path got attenuated by mainly by the change in slope.
However, to be on the safer side, it can be concluded that any check dam to arrest the flow and digest
initial thrust of the debris impact should be more than this height for this particular debris flow. This
study shows that rough estimation of heights of check dams for similarly vulnerable slopes can be
done by the development of such models in a simple but fast methodology. Spatial variation of
velocity and momentum of such flows can provide vital inputs to develop the design and extent of
remedial measures.
For further refinement of modeled outputs, influences of side-wise mass contribution, en-route
erosion, an influence of rheology and pore-pressure, relationship between discontinuity vis-à-vis
topography should be considered. The actual outputs can still be on the higher side as the model does
not include side-channel contribution and en-route erosion. Moreover, simulation output is required
to be verified with the previously modeled event as a part of validation strategy. In this context, the
input parameters are important because these parameters would affect the simulation results.  Rather
validation was carried out on collected field data in terms of their shear strength parameters and flow
characteristics. In this regard to get real field data, it is always recommended to collect such data at
the earliest after an event.

**Acknowledgement:**
Contributions of students and JRFs like Shobhana, Sweta and Gopal in different phases have helped
immensely to shape the paper. Organizational support and overall guidance provided by Dr. A.
Senthil Kumar, Director, IIRS and Dr. SPS Kushwaha, Ex-Dean (Academics) are also duly
acknowledged. Helps received from former Director of IIRS is placed on record. SLC and PKC is
thankful to Indian Space Research Organization, Department of Space, Government of India for the
financial support provided in TDP project.





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

**Table Captions:**
Table 1 Satellite data types and its sources
**Figure Captions:**
Figure. 1 Location map of the study area (Source: Astrium, May 3, 2016, © Google Earth). (Inset:
study area shown in Indian map)
Figure. 2 Geomorphological map (1:50000) of a part of Pune District, Maharashtra (Source: Bhuvan,
NRSC). Black and white line represent Pune district boundary and major road network respectively.
Figure 3.Filed photograph and satellite imagery. (a) Panoramic view of the Malin Landslide
(Photograph taken on September, 2015). Field length of photograph = 250m; (b) SPOT Image, Apr,
03, 2015 (© Google Earth); (c) Standard FCC of LISS IV, Jan 8, 2014 (RGB:321), Resourcest-2.
Black circle highlights Malin village.
Figure 4. (a) Subset of DEM of Malin area showing source area (in violet) and area of influence
(inside green boundary) of debris flow; (b) Elevation map Malin area
Figure. 5 Spatial variation of vital flow parameters of the debris flow model. (a) Momentum; (b)
pressure; (c) velocity and (d) height.







Fig. 1 Location map of the study area (Source : Astrium, May 3, 2016, © Google Earth).







Fig. 2 Geomorphological map (1:50000) of a part of Pune District, Maharashtra (Source: Bhuvan, NRSC). Black and white line represent Pune district boundary and major road network respectively.

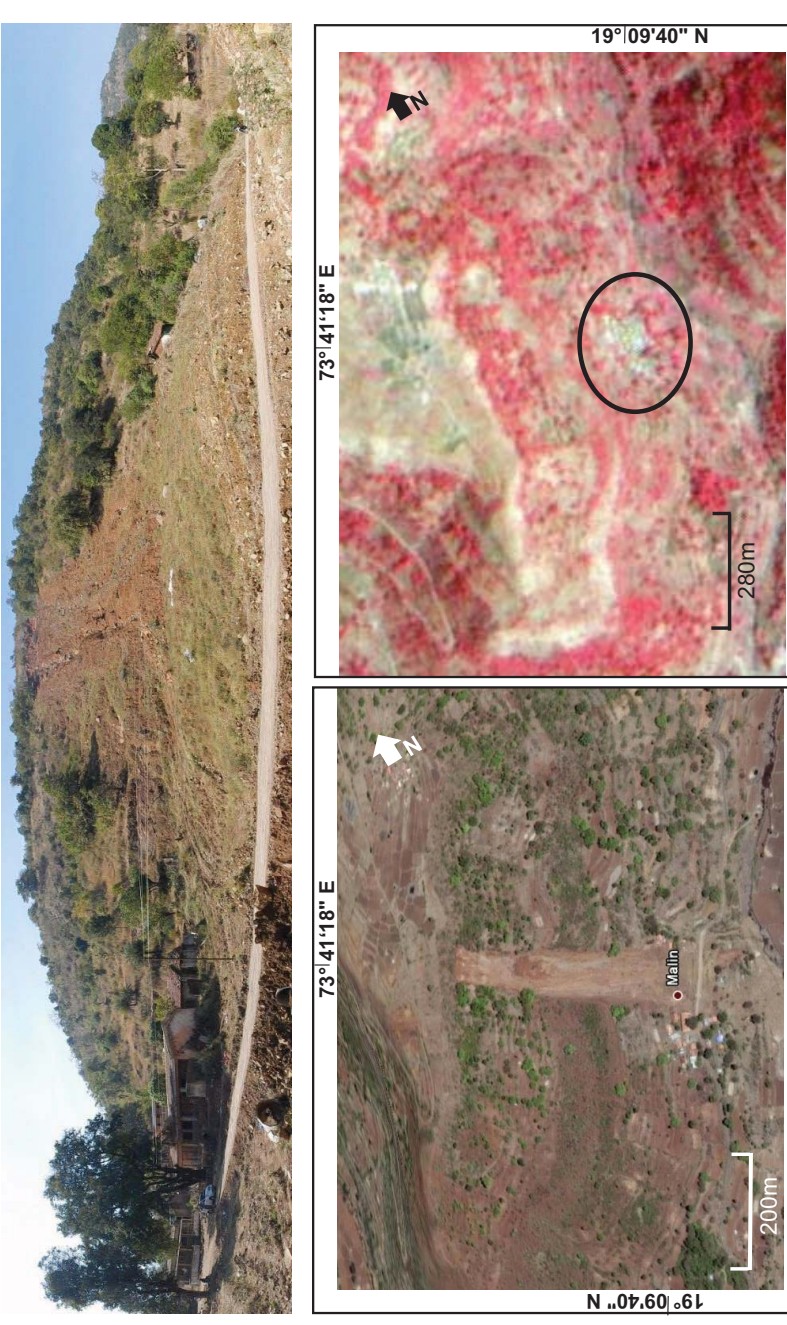

Fig. 3. Filed photograph and satellite imegray. (a) Panoramic view of the Malin Landslide (Photograph taken on September, 2015). Field length of photograph = 250m ; (b) SPOT Image, Apr, 03, 2015 (© Google Earth); (c) Standard FCC of LISS IV, Jan 8, 2014 (RGB:321), Resourcest-2. Black circle highlights Malin village.




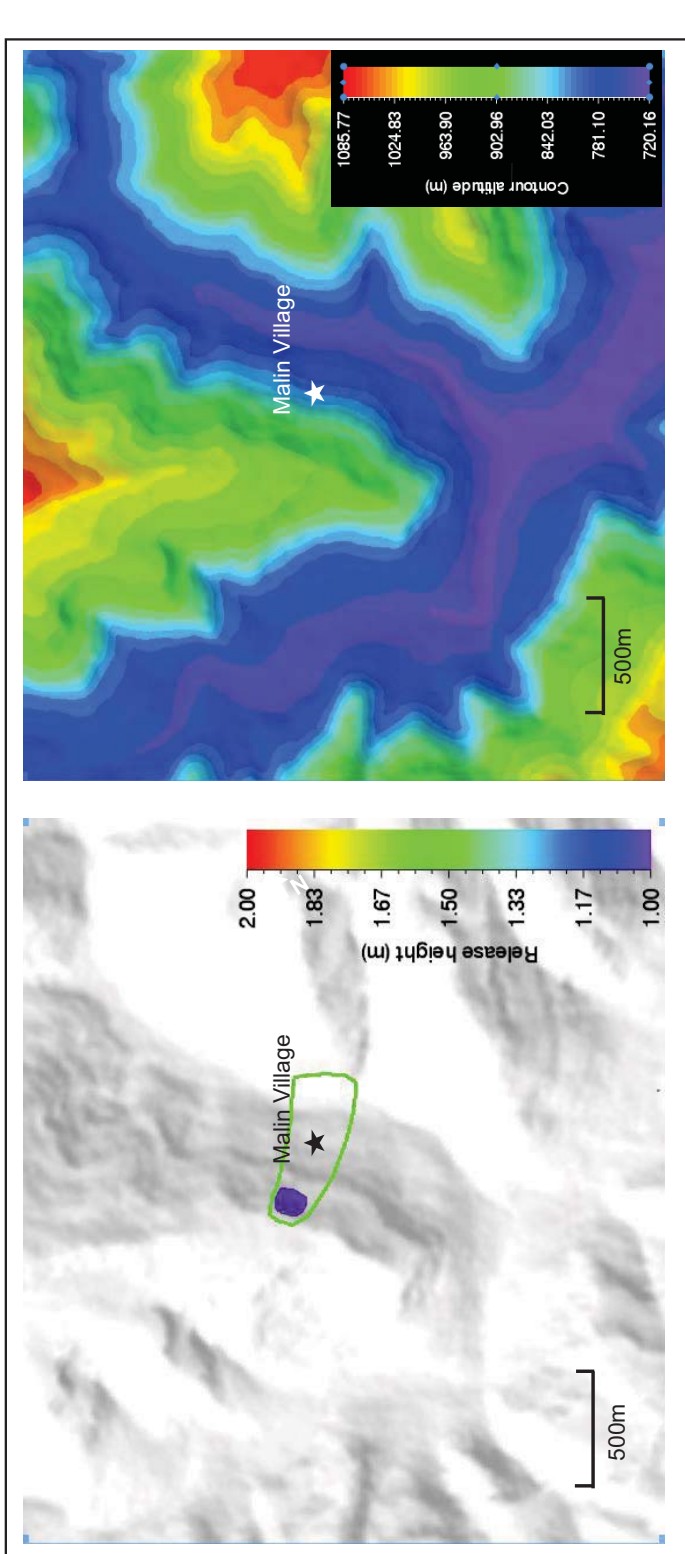

Fig. 4 (a) Subset of DEM of malin area showing source area (in violet) and area of influence (inside green boundary); (b) Elevation map Malin area.

Fig. 5 Spatial variation of vital flow parameters of the debris flow model. (a) Momentum (b) pressure (c) velocity and (d) height



Table 1. Satellite data types and its sources

| Data Type | Source |
|---|---|
| LISS-III, 02.12.2011 , 23.5 m resolution, 4 bands | Bhuvan, ISRO Geoportal |
| LISS-IV, 08.01.2011 (Pre-event) and 01.02.2015 (Post event), 5.2 m resolution, 4 bands | National Remote Sensing Centre (NRSC), ISRO |
| Cartisat-1, 06.04.2015 (Post event), 2.5 m, PAN | |
| Cartosat-1 stereo pair derived DEM, 10 m resolution (3$^{rd}$ March, 2011 ) | Bhuvan, ISRO Geoportal |
| SPOT satellite image | Google Earth |
| SRTM DEM, 90m | USGS Earth Explorer |
| Topographical map (E43B/12) | Survey of India |
| Geological information | Reports of GSI and published papers |