# Peer review of "3-Dimensional modeling of 2014-Malin Landslide, Maharashtra using satellite-derived data: A quantitative approach to numerical simulation technique 2 Shovan Lal Chattoraj1, Prashant K. Champati ray1, Sudhakar Pardeshi2, Vikram Gupta3, Yateesh 3 Keth"

_Natural Hazards and Earth System Sciences, 2017_

## Referee Comment (RC1) · Anonymous Referee #1 · 31 Jan 2018

The authors present a modelling study on a landslide occurred in south India. Even though the topic of the study is certainly suitable for the audience of Natural Hazards and Earth System Sciences, the overall quality of the paper in my opinion does not reach the journal standards and I think it should be rejected. Following, some broad comments about the scientific soundness of the research.

Broad comments

1. In the abstract "high-resolution DEM" are introduced. However, the subchapter 4.1 that should have described the quality, accuracy and error filtering of the multi-temporal DEM data is extremely short and providing scarce information about data and data handling. Moreover, I do have reserves about whether it is possible to define a 10 m cell DEM "high resolution", especially if the study area is not so large.

2. When speaking of multi-temporal DEM analysis I would have expected to find a map describing the thickness of the deposit and source area that would help me to compare the numerical model results with the actual phenomenon. In paragraph 4.2, you describe the depletion zone as "1 m to 1.2 m thick". Did you derived these values from the 10 m cell in the multi-temporal DoD? Did you estimate the errors? Also, data about the thickness of the deposit are missing.

3. I would have defined your landslide as a debris avalanche following the Hungr, O. (2001). A review of the classification of landslides of the flow type classification.

4. the equation in line 151 refers to the Voellmy rheology without cohesion. Nevertheless, your simulation considers cohesion and a very large one (100 kPa). The recommended values in the RAMMS online repository are the following: Avalanche, dry snow: 0 - 100 Pa; Avalanche, wet snow: 100 - 300 Pa; Debris Flow: 0 - 2000 Pa, so your selected values are two orders of magnitude larger than the higher limit recommended. This should be at least discussed

5. The statement that "the angle of internal resistance .. can be determined in laboratory" should be also discussed. This concept in fact conflicts with large literature that describes the difficulties of "scaling down" in laboratory the mechanics of an actual landslide. This has been expressed for example by Iverson, R. M. (1997). The physics of debris flows. Reviews of Geophysics, 35(3), 245, or by the number of large-scales experiments that are carried out by our colleagues all over the world or the necessity of back-analysis. This is therefore a matter that should be addressed and it could also

help expand the too short discussion

6. Do you think that your semi-consolidated samples of material tested in a shear test apparatus are representative of the loose material that collapsed during the event? How large is the fine fraction in your samples? Did you perform soil grading tests? The cohesion (around 100 kPa) is very large for a recent debris avalanche deposit and it is between what I would expect for a highly consolidated clay mixture or a weak rock mass.

Specific comments - The scale of figure 2 is too large and not of much use in the overall economy of the paper

---

## Referee Comment (RC2) · Anonymous Referee #2 · 12 Feb 2018

Dear authors, dear editor, The paper "3-Dimensional modeling of 2014-Malin Landslide, Maharashtra using satellite derived data: A quantitative approach by numerical simulation technique" presents a case study of propagation modelling based on the debris flow of Malin 2014. The text is clearly written and easy to understand, however the main problems is that there is no innovative content. In the present form, it looks more like a report for a local survey then a scientific paper. Several elements about input data are missing, some assumptions about the representativeness of lab values should be discussed. I do not think that this paper can be used by somebody working

outside from the specific case of Malin, then I propose not to consider this contribution for further publication.

Here are some points that should be significantly improved if the authors want to re-submit their work:

- The goals expressed in the introduction (lines 60-70) are quite far from the work actually achieved. These goals should be reformulated. For instance it could refocused on the assessment of using lab measurements to constrain Voellmy's parameters for this kind of debris movement.

- Input data should be better described. For instance there is no proper discussion of the DEM quality built for this project, and Fig4 doesn't allow to have a feeling of how it looks. Line 204 "representative samples" for shear tests: how many, how do you know they are representative, what is the variability of these measurements, where are these measurments ?

- The authors state that the internal friction required by the model can be directly extracted from shear tests on samples. That's by far not so simple: scaling problems, representativeness of samples, grain size effect, etc. The role of cohesion is not clear (it doesn't not appear in line 151). All this part about lab measurement is treated too superficially for a scientific contribution.

―――――――――――――――――――――